# Latent class analysis of substance use typologies associated with mental and sexual health outcomes among sexual and gender minority youth

Tara Carney [1,2]*, Seul Ki Choi[3], Rob Stephenson[4], Jose A. Bauermeister[3], Adam W. Carrico[5]

1 Mental Health, Alcohol, Substance Use and Tobacco Drug Research Unit, South African Medical Research Council, Tygerberg, South Africa, 2 Department of Psychiatry and Mental Health, Division of Addiction Psychiatry, University of Cape Town, Groote Schuur Hospital, Observatory, Cape Town, South Africa, 3 Department of Family and Community Health, School of Nursing, University of Pennsylvania, Philadelphia, Pennsylvania, United States of America, 4 School of Nursing, University of Michigan, Ann Arbor, Michigan, United States of America, 5 Stempel College of Public Health and Social Work, Florida International University, Miami, Florida, United States of America

* tara.carney@mrc.ac.za

**Data Availability Statement:** All relevant data are within the paper and its Supporting Information files.

## Abstract

Little is known about sexual and gender minority youth (SGMY) who have sex with men's unique patterns of substance use, even though they are at risk for substance use and adverse mental and other health outcomes. We used latent class analysis to examine typologies of substance use and multinomial logistic regression to investigate mental health outcomes (depression and anxiety) and HIV/STI testing correlates associated with different classes of substance use in a sample of SGMY who have sex with men in the USA and use substances (n = 414) who participated in an online survey. The average age was 22.50 years old (SD = 3.22). A four-class solution was identified representing: 'depressant and stimulant use' (3.4%), 'high polysubstance use' (4.6%), 'low substance use with moderate cannabis use' (79.2%), and 'high cannabis, stimulant and alcohol use' (12.8%). Membership to a specified substance use class varied by age, previous arrest, gender identity, anxiety, and lifetime HIV testing. Multivariate logistic regression results indicated that participants in the high polysubstance use (AOR = 5.48, 95% CI 1.51, 19.97) and high cannabis use class (AOR = 3.87, 95% CI 1.25, 11.94) were significantly more likely than those in the low substance with moderate cannabis use class to report previous arrest. Those in the high polysubstance use class were also significantly less likely to have been tested for HIV than those in the low substance use with moderate cannabis use class (AOR = 0.21, 95% CI 0.05, 0.93). Findings will guide the development and implementation of tailored approaches to addressing the intersection of substance use and HIV risk among SGMY.

**Funding:** The primary funder (R Stephenson) of this study is the United States National Institute on Drug Abuse (NIDA) (5R01DA041032-02); the study protocol underwent peer-review by the funding body. The content is solely the work of the authors and does not necessarily reflect the views of NIDA, who did not have a role in the study design, data collection and analysis plans, decision to publish, or preparation of the manuscript.

**Competing interests:** The authors have declared that no competing interests exist.

## Introduction

Sexual and gender minority youth (SGMY) is a term that includes sexual minority youth (including lesbian, gay, bisexual, queer) and gender minority youth (including transgender, genderqueer) as well as youth who are uncertain of their sexual orientation and/or gender identity.

Substance use is highly prevalent amongst SGMY who have sex with men. For example, the 2019 national HIV Behavior Surveillance Survey conducted in the United States of America (USA) found that over half of men who had sex with men (MSM) in the sample used illegal drugs, although this did not focus on young sexual minority youth (SMY) who have sex with men [1]. Studies have shown that SMY globally [2], including in the USA [3,4], are more likely to use substances in comparison to their heterosexual counterparts. Data collected from the national Youth Risk Behavior Survey from 2005–2017 found that gay (AOR = 4.33, 95% CI 3.36, 5.57), bisexual (AOR = 3.36, 95% CI: 2.68–4.20) and young men who were uncertain about their sexual orientation (AOR = 3.18, 95% 2.63, 3.84) had increased odds for drug use [5]. In addition, research shows early initiation of substance use among SMY which has been found to progress into problem substance use or a substance use disorder during adulthood [6–8]. For example, among a sample of male high school students, binge drinking prevalence was higher among SMY who have sex with men (26%) in comparison to their heterosexual peers (19%) [9]. A similar pattern has been found among gender minority youth (GMY), who have also been found to have up to four times higher substance use prevalence than cisgender youth [10].

Recent studies indicate that polysubstance use is increasingly common among SGMY, with up to 53% using more than one drug [11,12]. It is difficult to estimate prevalence due to convenience sampling utilized in many studies, but it is critical to examine the typologies of substance use among groups of youth that have typically faced multiple types of discrimination to understand consequences of polysubstance use and guide the development and implementation of tailored intervention approaches to address this high priority population's comprehensive care needs.

The minority stress theory developed by Meyer [13] suggests that sexual and gender minorities, including SGMY, experience unique sexuality and gender-based stressors such as abuse, violence and other forms of discrimination because of their stigmatized social status [14], as well as self-stigma and concealment of sexual orientation or gender identity [15]. These unique, often lifelong, stressors increase their risk for physical and mental health problems, including substance use [16]. Because SGMY are more likely to experience social marginalization, violence and victimization, they can develop mental health comorbidities like anxiety, depression, posttraumatic stress disorder and suicidality which are in turn are associated with negative health outcomes including sexual health outcomes [11,17,18].

This is important for SMY who have sex with men, who are particularly at risk for HIV and other sexually transmitted infections [11,19,20]. Likewise, for GMY, a recent study also estimated HIV prevalence among transgender youth at over 30% [21]. Studies have also found high levels of substance use that aim to enhance sex and increased STI and HIV risk [22,23]. Substance use is associated with condomless anal sex in SGMY who have sex with men [21,24], especially club drugs [24], an increase in number of sexual partners, and sex with a partner living with HIV [23]. Despite guidelines that sexually active key populations receive HIV testing every six months to facilitate entry into the HIV prevention and care [25,26] continuum, studies have found that heavy alcohol use is a barrier to HIV testing among SMY who have sex with men [27,28], but research on the role of other drugs is lacking. While the knowledge rates of HIV testing among GMY including transgender and non-binary youth are

limited, a recent study indicates that levels are testing are generally low, especially if gender- and age-appropriate services are not available [29].

While substance use in general is associated with increased HIV risk, less is known about *how* typologie*s* of drug use may affect HIV risk, and entry into the HIV continuum of care. The present study aims to identify substance use typologies among a sample of SGMY who use substances in Detroit, Michigan in the USA. It also aims to examine whether the identified substance use classes are associated with demographic variables, mental health, and sexual health outcomes.

## Methods

### Study design and participants

We utilized baseline data (n = 414) from a 4-arm factorial randomized controlled trial to evaluate the effectiveness of a brief intervention targeting substance use reduction and increased engagement in HIV prevention among SGMY who engage in sex with men in Detroit, Michigan. Recruitment took place between April 2017 and September 2019 and study details are described elsewhere [30].

Eligibility criteria for the study included: 1) between 15 and 29 years old; 2) gender identity: self-identify as cisgender man or transgender man or woman; 3) sexual practices: report oral or anal sex with a man in the last 6 months; 4) live in the Detroit Metro area; 5) have unknown or negative HIV status; and 6) report substance use in the previous three months.

### Data collection

Recruitment methods entailed the use of Web-based advertisements on social media websites and apps such as Facebook, Grindr, and community-based outreach in local venues and specific events. All potential participants completed informed consent and a baseline survey online (the focus of this paper) before enrolling into the study. All procedures performed in studies involving human participants were in accordance with the ethical standards of the institutional research committee and with the 1964 Helsinki declaration and its later amendments or comparable ethical standards. Ethics approval for this study was granted by the Institutional Review Board at the University of Michigan.

### Measures

**Latent class indicators.** For the current study, we used the ASSIST (Alcohol, Smoking and Substance Involvement Screening Test), a validated measure developed by the World Health Organization (WHO) to screen for the presence of possible alcohol and other substance use disorders [31]. A range of substances were assessed, including nonmedical cannabis, amyl nitrate (poppers), cocaine, methamphetamine, gamma-hydroxybutyrate (GHB), prescribed stimulants, prescription sedatives (such as Xanax, Valium), Ecstasy, ketamine, heroin, opioid medications, and hallucinogens (psilocybin mushrooms, lysergic acid diethylamide [LSD], PCP). For this analysis, we focused on any use in the past three months.

For alcohol use, we used the WHO's widely used and validated Alcohol Use Disorders Identification Test (AUDIT), a 10-item screening questionnaire that includes questions on amount and frequency of alcohol use, alcohol dependence and problems caused by alcohol use [32]. For men, an AUDIT score of more than 8 indicates hazardous or harmful use.

**Exposure variables.** Sociodemographic factors included age, racial and ethnic categories, sexual identity and gender identity. Sexual identity was categorized as gay, bisexual, and other, and gender identity as cisgender (gender identity matches sex assigned at birth), transgender,

or non-binary. Racial and ethnic categories were combined, and categorized into White, Black, Hispanic/Latino, other race, or multi-racial. Other demographic factors included level of education, employment, housing, previous abuse by police (defined as unfairly stopped, searched, questioned, physically threatened or abused by the police), and lifetime arrest.

Sexual health variables included previous HIV testing measured by lifetime number of tests, lifetime diagnosis of any sexually transmitted infections (STIs), and pre-exposure prophylaxis PrEP) awareness and use.

Anxiety was assessed using the Generalized Anxiety Disorder Questionnaire (GAD-7), a tool shown to have good reliability and validity [33], with scores categorized as mild (5–9), moderate (10–14) and severe (15 and higher) anxiety. Depression was measured using the Centre of Epidemiological Studies Depression Scale (CESD-10), a short screener with a cut-off score of 10 or higher indicating the presence of clinically-relevant depressive symptoms [34].

## Data analysis

Mplus statistical modelling software version 8.4 was used to conduct latent class analysis (LCA) [35], to identify underlying "classes" or subgroups of participants through patterns of covariance in the data structure. To select the number of latent classes in our LCA model, several information criteria were reviewed including Akaike Information Criterion [AIC], Bayesian Information Criterion [11] and entropy. Smaller AIC and BIC values are a better fit to the true model, and entropy values nearing 1 suggest a greater class separation. We also used a bootstrap likelihood ratio test (BLRT) to estimate the distribution of the log likelihood difference test statistic. Significant p-value between the $k$-1 class model and the $k$ class model indicates a significant improvement in model fit. Finally, we considered empirical and theoretical relevance and sample size of the class when we determine the number of latent classes. In Mplus, missing data is handled using a technique called full information maximum likelihood (FIML) estimation. Mplus leverages FIML estimation to optimize the utilization of available data and generate robust estimates of model parameters, even in the presence of missing data points.

Bivariate multinomial logistic regression models were then used to calculate odds ratios and 95% confidence intervals for the above-mentioned correlates of latent class membership (the outcome variable). The adjusted multinomial logistic regression models consist of all the variables used in the bivariate multinomial logistic regression models, except for the racial and ethnic categories. In the adjusted model, White and Hispanic/Latino dummy variables were included in the adjusted model due to the risk of overfitting and unbalanced cell sizes. Vermunt's 3-step approach was used in covariate analysis to take classification uncertainty into consideration by fitting the mixture model without including any external covariate variables and assigning individuals into classes based on their best posterior probability.

## Results

Participants were 22.50 years old (SD = 3.22) on average. The majority self-identified as gay (n = 270, 65.2%) and cisgender men (n = 331, 80.0%). More than three-quarters had never been arrested (n = 329, 79.5%). Approximately one-third respectively were either employed full-time (n = 157, 37.9%) or part-time (n = 129, 31.2%), with a further 23.4% (n = 97) unemployed, or 7.49% (n = 31) unable to work. More than two-thirds of the sample self-identified as White (n = 264, 63.8%).

Over two-thirds of participants had ever been tested for HIV (n = 302, 73.0%), on average 3.61 times (SD = 6.52). In addition, 69.57% (n = 288) tested for sexually transmitted infections (STI) in their lifetime and 12.80% (n = 53) had received a positive diagnosis. While most

participants were aware of PrEP (n = 274, 66.2%), only 6.3% (n = 26) had ever taken PrEP; 7.97% (n = 33) reported currently being on PrEP. Mental health measures indicated that a high proportion scored above the cut-off point for depression (n = 252, 61.2%) and minimal (n = 120, 29.0%), mild (n = 119, 28.7%), moderate (n = 77, 18.6%) or severe (n = 98, 23.6%) anxiety.

## Substance use prevalence

Cannabis was the most commonly reported substance used in the past three months (n = 284, 68.6%). Approximately a quarter of participants reported using stimulants including cocaine, methamphetamine and prescription stimulants (n = 98, 23.7%). Similar proportions reported use of amyl nitrite (n = 65, 15.7%), hallucinogens (n = 58, 14.0%) and sedatives (n = 50, p = 12.1%), followed by other club drugs such as Ecstasy, GHB and ketamine (n = 36, 8.7%) and prescription and illegal opioids (n = 20, 4.8%). 21.5% (n = 89) reported no drug use, 42.3% (n = 175) reported the use of only one drug, and 36.2% (n = 150) used more than one substance in the past three months. Over a third screened positive for hazardous or harmful alcohol use (n = 147, 35.5%).

## Class enumeration

We iteratively compared models with increasing numbers of class solutions. Table 1 shows similar fit statistics for 2, 3, 4 and 5 class solutions. Given that traditional fit indices may not uniformly point to a single model specification, we selected a final number of latent classes by considering empirical and theoretical perspectives and interpretability of the results. Thus, a 4-class solution model was selected in our study. Although the BIC and entropy values did not support our final model specification, but the lowest AIC and the significant p-value for bootstrap likelihood ratio tests (p<0.0001) support our 4-class solution.

Fig 1 shows the results of the 4-class solution, including the prevalence of substance use indicators by class. Class 1 represents 3.4% (n = 14) of the included sample and was characterized by "depressant and stimulant use". Specifically, all members used some kind of depressant (sedatives) (100%); or opioids (46.7%) and a large proportion reported high-risk alcohol use (67.9%). Stimulant (73.1%) although somewhat less so than in Class 2.

Class 2 is categorized as "high polysubstance use" which constitutes 4.6% (n = 19) of the sample. All (100%) of the participants in this class reported the use of stimulants, cannabis, *and* hallucinogens in the past 3-months, with almost all also reporting the use of sedatives (93.7%). Moreover, they reported moderate levels of high-risk alcohol (62.9%), opioids (55.3%), club drugs (70.1%), and amyl nitrite (56.7%).

Class 3 is considered the "low substance use with moderate cannabis use" class, which makes up 79.2% (n = 328) of the sample. This class showed low substance use levels except moderate cannabis use (62.1%), as well as the lowest proportion of high-risk alcohol use (25.9%).

**Table 1. Latent class analysis fit statistics.**

| N Classes | AIC | BIC | aBIC | Entropy | -2Log-likelihhood | BLRT Log-likelihood | BLRT p-value |
|---|---|---|---|---|---|---|---|
| 2 class | 2689.136 | 2757.575 | 2703.630 | 0.860 | 2,655.136 | -1456.155 | <0.0001 |
| 3 class | 2672.392 | 2777.064 | 2694.560 | 0.798 | 2,620.392 | -1327.568 | <0.0001 |
| 4 class | 2660.430 | 2801.335 | 2690.272 | 0.846 | 2,590.43 | -1310.196 | <0.0001 |
| 5 class | 2665.216 | 2842.354 | 2702.732 | 0.855 | 2,577.216 | -1295.215 | 0.4286 |

**Note:** AIC: Akaike's Information Criterion; BIC: Bayesian Information Criterion; aBIC: Sample size adjusted BIC; BLRT: Bootstrap likelihood ratio test.

Probabilities of individual substance use by substance use latent classes

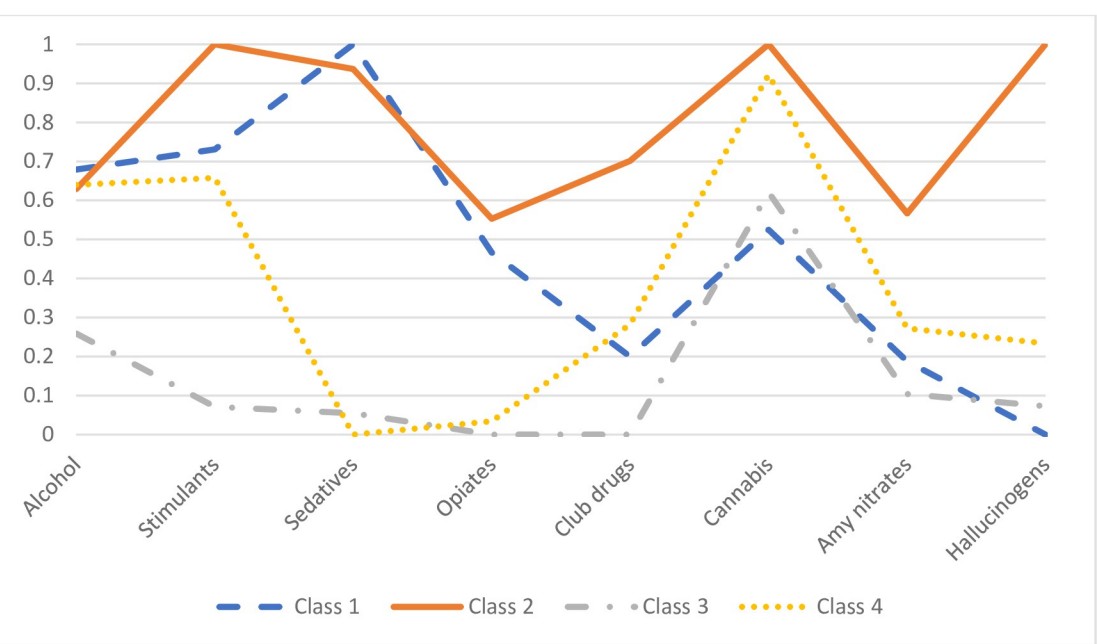

**Fig 1. Substance use by latent class.**

Class 4, classified as "high cannabis, stimulant and alcohol use" class, accounted for 12.8% (n = 53) of the sample. Participants in this class had the highest probability of cannabis use (92.2%), followed by stimulant use (65.8%) and high-risk alcohol use (64.0%). Club drug (28.2%), amyl nitrate (27.2%) and hallucinogen (23.4%) use were reasonably low, with opioid use being very low (3.4%), and no sedative use (0%) was reported.

## Factors associated with substance use

Table 2 presents sociodemographic characteristics, mental health and HIV/STI behaviors associated with these classes. In terms of socio-demographic factors, an ANOVA test of means revealed significant differences in age (f (3, 2.77); p = 0.04), with participants in Class 2 (high polysubstance use) having the youngest mean age (21.9, SD = 2.88). Chi-square tests of association found a significant association between sexual orientation and substance use class, ($\chi 2$ (3) = 8.32; p = 0.04) with bisexual participants being most likely to be in Class 2. Participants in Class 1 (depressant and stimulant use) were significantly more likely to have been previously arrested ($\chi 2$ (3) = 21.04; p<0.001) or abused by police ($\chi 2$ (3) = 9.91; p = 0.02). Measures of mental health outcomes were also significantly associated with substance use class. Severe anxiety was most prevalent in Class 1 and 2 ($\chi 2$ (9) = 29.20; p<0.001). Similarly, having a score which indicated depression was more likely for participants in Class 1 and 2 ($\chi 2$ (3) = 11.47; p = 0.01). A significant association between substance use class and lifetime HIV testing was observed, with participants in Class 2 being least likely to have been tested for HIV ($\chi 2$ (3) = 11.39; p = 0.001).

## Unadjusted and adjusted multinomial logistic regression models of substance latent classes

Table 3 presents the unadjusted and adjusted models associated with substance use classes as outcomes, with Class 3 (cannabis and alcohol use) as the referent class. In the adjusted model,

**Table 2. Association between substance use latent classes, socio-demographics, and health outcomes among sexual and gender minority youth in the Detroit Metro area.**

| | TOTAL (n = 414) | Class 1 (n = 14) | Class 2 (n = 19) | Class 3 (n = 328) | Class4 (n = 53) | p- value |
|---|---|---|---|---|---|---|
| *Sociodemographic Variables* | | | | | | |
| Age | 22.5 (3.2) | 23.3 (3.9) | 21.9 (2.9) | 22.3 (3.2) | 23.6 (3.2) | **0.04** |
| Racial and Ethnic Categories | | | | | | |
| *White/Caucasian* | 264 (63.8%) | 9 (64.3%) | 15 (79.0%) | 203 (61.9%) | 37 (69.8%) | 0.43 |
| *Black* | 54 (13.0%) | 2 (14.3%) | 0 (0%) | 49 (14.9%) | 3 (5.7%) | |
| *Mixed* | 30 (7.3%) | 1 (7.1%) | 3 (15.8%) | 22 (9.2%) | 4 (7.6%) | |
| *Other* | 26 (6.3%) | 0 (0%) | 1 (5.3%) | 22 (9.2%) | 3 (5.7%) | |
| *Hispanic* | 40 (9.7%) | 2 (14.3%) | 0 (0%) | 32 (9.8%) | 6 (11.3%) | |
| Gender | | | | | | |
| *Cismen* | 331 (80.0%) | 14 (100%) | 15 (79.0%) | 257 (78.4%) | 45 (84.9%) | 0.60 |
| *Transmen* | 46 (11.1%) | 0 (0%) | 3 (15.8%) | 38 (11.6%) | 5 (9.4%) | |
| *Transwomen* | 13 (3.1%) | 0 (0%) | 0 (0%) | 13 (4.0%) | 0 (0%) | |
| *Nonbinary* | 24 (5.8%) | 0 (0%) | 1 (5.3%) | 20 (6.1%) | 3 (5.7%) | |
| Sexuality | | | | | | |
| *Gay* | 270 (65.2%) | 13 (92.9%) | 11 (57.9%) | 209 (63.7%) | 37 (69.8%) | **0.02** |
| *Bisexual* | 77 (18.6%) | 1 (7.1%) | 8 (42.1%) | 59 (18.0%) | 9 (17.0%) | |
| *Other* | 67 (16.2%) | 0 (0%) | 0 (0%) | 60 (18.3%) | 7 (13.2%) | |
| Education | | | | | | |
| *Some high* | 26 (6.3%) | 1 (7%) | 2 (10.5%) | 20 (6.1%) | 3 (5.7%) | 0.38 |
| *High school grad/GED* | 93 (22.5%) | 2 (14.3%) | 2 (10.5%) | 82 (25.0%) | 7 (13.2%) | |
| *Some college >* | 295 (71.3%) | 11 (78.6%) | 15 (79.0%) | 226 (68.9%) | 43 (81.1%) | |
| Employment (full-time) | 157 (37.9%) | 8 (57.1%) | 6 (31.6%) | 121 (36.9%) | 22 (41.5%) | 0.40 |
| Housing | 261 (63.0%) | 11 (78.6%) | 11 (57.9%) | 204 (62.2%) | 35 (66.0%) | 0.58 |
| Previous arrest | 85 (20.5%) | 7 (50.0%) | 8 (42.1%) | 53 (16.2%) | 17 (32.1%) | <**0.01** |
| Abused by police | 83 (20.1%) | 7 (50.0%) | 5 (26.3%) | 64 (19.5%) | 7 (13.2%) | **0.02** |
| *Mental Health Outcomes* | | | | | | |
| Anxiety | | | | | | |
| *Minimal* | 120 (29.0%) | 0 (0%) | 3 (15.8%) | 102 (31.1%) | 15 (28.3%) | <**0.01** |
| *Mild* | 119 (28.7%) | 3 (21.4%) | 2 (10.5%) | 102 (31.1%) | 12 (22.6%) | |
| *Moderate* | 77 (18.6%) | 2 (14.3%) | 4 (21.1%) | 60 (18.3%) | 11 (20.8%) | |
| *Severe* | 98 (23.7%) | 9 (64.3%) | 10 (52.6%) | 64 (19.5%) | 15 (28.3%) | |
| Depression | 252 (61.2%) | 13 (92.9%) | 16 (84.2%) | 194 (59.5%) | 29 (54.7%) | <**0.01** |
| *HIV/STI Prevention Behaviours* | | | | | | |
| Lifetime HIV Testing | 302 (73.0%) | 12 (85.7%) | 8 (42.1%) | 240 (73.2%) | 42 (79.3%) | <**0.01** |
| Repeat HIV testing | 0 (1.0) | 0.04 (0.49) | -0.28 (0.43) | -0.02 (1.06) | 0.20 (0.84) | 0.30 |
| Lifetime STI testing | 288 (69.6%) | 10 (71.4%) | 9 (47.4%) | 230 (70.1%) | 39 (73.6%) | 0.39 |
| STI diagnosis | 53 (12.8%) | 2 (14.3%) | 0 (0%) | 39 (11.9%) | 12 (22.6%) | 0.05[a] |
| PrEP | | | | | | |
| *Unaware/Aware* | 355 (85.8%) | 13 (92.9%) | 17 (89.5%) | 283 (86.3%) | 42 (79.3%) | 0.69 |
| *Past use* | 26 (6.3%) | 0 (0%) | 1 (5.3%) | 19 (5.8%) | 6 (11.3%) | |
| *Current use* | 33 (8.0%) | 1 (7.1%) | 1 (5.3%) | 26 (7.9%) | 5 (9.4%) | |

[a] Please note this exact p-value is 0.0536, which is not considered significant.

**Table 3. Results of multinomial logistic regression models between substance use latent classes, socio-demographics, and health outcomes among sexual and gender minority youth in the Detroit Metro area.**

| | Class 1: Depressant and Stimulant Use n = 14 (3.4%) | | Class 2: High Polysubstance Use n = 19 (4.6%) | | Class 4: Cannabis, Stimulant and Harmful/Hazardous Alcohol Use n = 53 (12.8%) | |
|---|---|---|---|---|---|---|
| | OR (95% CI) | AOR (95% CI) | OR (95% CI) | AOR (95% CI) | OR (95% CI) | AOR (95% CI) |
| **Sociodemographic Variables** | | | | | | |
| Age | 1.13 (0.93; 1.37) | 1.01 (0.73, 1.38) | 0.97 (0.83; 1.12) | 0.85 (0.63, 1.14) | **1.17 (1.03, 1.31)** | **1.22 (1.02, 1.45)** |
| Race/Ethnicity | | | | | | |
| *White/Caucasian* | 1.15 (0.34, 3.92) | 1.82 (0.30, 10.99) | 2.40 (0.76, 7.64) | 1.37 (0.35, 5.43) | 1.54 (0.70, 3.40) | 1.93 (0.66, 5.63) |
| *Black* | 0.88 (0.17, 4.72) | - | - | - | 0.24 (0.04, 1.51) | - |
| *Mixed* | 1.09 (0.11, 10.47) | - | 2.67 (0.46, 10.16) | - | 1.15 (0.29, 4.61) | - |
| *Other* | - | - | 0.76 (0.09, 6.24) | - | 0.80 (0.17, 3.83) | - |
| *Hispanic* | 1.62 (0.31, 8.50) | 2.48 (0.18, 35.02) | - | - | 1.24 (0.40, 3.86) | 1.96 (0.36, 10.70) |
| Sexual identity | | | | | | |
| *Gay* | 11.79 (0.45, 306.06) | - | 0.80 (0.31, 2.09) | - | 1.43 (0.65, 3.17) | 1.60 (0.34, 7.64) |
| *Bisexual* | 0.30 (0.02, 3.74) | - | **3.43 (1.26, 8.87)** | - | 0.89 (0.34, 2.36) | 1.36 (0.25, 7.57) |
| Gender | | | | | | |
| *Cismen* | - | - | 1.07 (0.34, 3.39) | 0.84 (0.06, 12.44) | 1.75 (0.63, 4.92) | 0.94 (0.15, 5.82) |
| *transgender* | - | **10.62 (1.09, 103.29)** | 0.99 (0.27, 3.57) | 0.57 (0.03, 11.13) | 0.48 (0.13, 1.75) | 0.38 (0.04, 3.70) |
| Education | 1.39 (0.45, 4.24) | 1.12 (0.23, 5.52) | 1.21 (0.53, 2.77) | 2.32 (0.65, 8.27) | 1.71 (0.78, 3.72) | 1.05 (0.41, 2.68) |
| Housing | 2.50 (0.76, 8.28) | 3.55 (0.73, 17.25) | 0.80 (0.29, 2.22) | 1.05 (0.34, 3.31) | 1.29 (0.62, 2.68) | 0.80 (0.32, 1.97) |
| Employment (tttttime) | 2.47 (0.56, 10.84) | 2.65 (0.61, 11.57) | 0.85 (0.32, 2.21) | 1.28 (0.35, 4.71) | 1.24 (0.58, 2.67) | 1.45 (0.56, 3.72) |
| Previous arrest arresaaarreincarceration | **6.58 (1.96, 22.13)** | 3.43 (0.77, 15.37) | **4.30 (1.59, 11.63)** | **5.48 (1.51, 19.97)** | **3.01 (1.34, 7.02)** | **3.87 (1.25, 11.94)** |
| Police abuse | **4.48 (1.35, 14.62)** | 3.15 (0.64, 15.50) | 1.46 (0.50, 4.29) | 0.95 (0.25, 3.58) | 0.56 (0.19, 1.68) | 0.29 (0.07, 1.11) |
| **Mental Health Variables** | | | | | | |
| Anxiety | **3.14 (1.53, 6.48)** | **3.15 (1.27, 7.81)** | **2.07 (1.29, 3.84)** | **2.48 (1.19, 5.17)** | 1.25 (0.90, 1.91) | **2.03 (1.18, 3.50)** |
| Depression | 14.28 (0.47, 437.37) | 2.01 (0.10, 38.81) | **3.77 (1.02, 21.42)** | 1.37 (0.19, 9.94) | 0.78 (0.38, 2.03) | 0.31 (0.10, 1.03) |
| **HIV/STI prevention Behaviours** | | | | | | |
| HIV testing | 2.48 (0.43, 14.44) | 2.42 (0.29, 20.61) | **0.27 (0.10, 0.71)** | **0.21 (0.05, 0.93)** | 1.58 (0.63, 3.93) | 0.74 (0.23, 2.34) |
| Repeat HIV testing | 1.37 (0.62, 3.05) | 1.12 (0.42, 2.99) | 0.40 (0.10, 2.34) | 0.98 (0.26, 3.66) | **1.85 (1.06, 3.86)** | 1.17 (0.69, 2.00) |
| STI diagnosis | 1.39 (0.26, 7.32) | 1.57 (0.16, 15.11) | - | - | **2.62 (1.06, 6.52)** | 2.92 (0.86, 9.86) |
| PrEP Continuum | 0.75 (0.20, 2.84) | 0.49 (0.09, 2.67) | 0.82 (0.31, 2.18) | 3.03 (0.76, 12.14 12.14) | 1.32 (0.76, 2.30) | 0.89 (0.40, 1.97) |

OR = unadjusted odds ratio; AOR = adjusted odds ratio; Class 3 (low substance use) is a referent group. Due to low cell sizes, some results were unable to calculate odds ratio.

in terms of sociodemographic variables, age was significantly associated with substance use class, with participants in Class 4 (cannabis, stimulant and alcohol use) being 17% more likely to be older than those in the referent class (OR = 1.17, 95% CI = 1.03, 1.31). After adjusting for other factors in unadjusted models, age was still significantly associated with Class 4 (AOR = 1.22, 95% CI = 1.02, 1.45). In terms of sexual identity, the odds of being in class 2 (high polysubstance use) compared to class 3 were greater among bisexual participants (OR = 3.43, 95%, CI = 1.26, 8.87), but the association was no longer significant after controlling for other factors. In the adjusted model, transgender youth were significantly more likely to belong to Class 1 (depressant and stimulant use) (AOR = 10.62, 95% CI = 1.09, 103.29).

In the unadjusted model, those with high depressant and stimulant use (Class 1) (OR = 6.58, 95% CI = 1.96, 22.13), polysubstance use (Class 2) (OR = 4.30, 95% CI = 1.59, 11.63) and cannabis, stimulant and alcohol use (Class 4) (OR = 3.01, 95% CI = 1.34, 7.02) were

significantly more likely to have been arrested compared to those with predominantly cannabis and alcohol use (Class 3). In addition, those who reported abuse by police were significantly more likely to be in Class 1 OR = 4.48, 95% CI = 1.35, 14.62) compared to Class 3. In the adjusted model, previous arrest remained significantly associated with Class 2 (AOR = 5.48, 95% CI = 1.51, 19.97) and Class 4 (AOR = 3.87, 95% CI = 1.25, 11.94).

For the mental health outcomes, the unadjusted model also indicates that anxiety was significantly higher in Class 1 (OR = 3.14, 95% CI = 1.53, 6.48) and Class 2 (OR = 2.07, 95% CI = 1.29, 3.84) compared to Class 3. When we adjusted for other factors, anxiety was still significantly associated with Class 1 (AOR = 3.15, 95% CI = 1.27, 7.81), 2 (AOR = 2.48, 95% CI = 1.19, 5.17), and 4 (AOR = 2.03, 95% CI = 1.18, 3.50) compared to Class 3. Participants who were in the high polysubstance use class (Class 2) were also significantly more likely to report symptoms of depression (OR = 3.77, 95% CI = 1.02, 21.42) than the referent class in the unadjusted model. This association was no longer significant in the adjusted model.

Certain sexual health-seeking behaviors were significant in the unadjusted model. Participants who used multiple substances (Class 2) were significantly less likely to ever test for HIV than those in Class 3 in the unadjusted (OR = 0.27, 95% CI = 0.10, 0.71) and adjusted model (AOR = 0.21, 95% CI = 0.05, 0.93). In the unadjusted model, repeat HIV testing (OR = 1.85, 95% CI = 1.06, 3.86) and having a previous diagnosis of an STI (OR = 2.62, 95% CI = 1.06, 6.52) were both significantly associated with Class 4 (cannabis, stimulant, alcohol use) in comparison to Class 3. In the adjusted model, neither of these were significant.

## Discussion

The current study findings provide prevalence estimates of recent multiple substance use in a sample of SGMY who have sex with men. Cannabis was the most prevalent substance (68.6%) used, followed by hazardous/harmful alcohol (35.5%) and stimulant use (23.7%). Distinct typologies of substance use were found, and were correlated with age, gender identity, previous arrest, mental health symptoms, and lifetime HIV testing. Overall, results underscore substantial heterogeneity in substance use typologies among this sample of SGMY that may elucidate the need for tailored intervention approaches targeting the intersection of substance use, mental health symptoms and HIV risk in this high priority population.

According to LCA results, substance use patterns fit into one of four discrete classes: 1) depressant and stimulant use, 2) high polysubstance use, 3) low substance use and moderate cannabis use and 4) high cannabis, stimulant and alcohol use. While the majority of participants were classified as having low substance use, moderate cannabis and some hazardous/ harmful alcohol use was prevalent even among this group. The second most common classification was the high cannabis, stimulant and alcohol use group, since a high level of stimulant use coupled with hazardous/harmful alcohol use were also found. This study adds to our understanding that SGMY may have different substance use patterns to other groups of MSM and gender minority populations, including higher levels of stimulant and opioid use [27,36,37]. However, high prevalence of cannabis and alcohol match the findings from large youth surveys conducted in the USA [38,39]. Further, it also supports minority stress as predictive of substance use among SGYM [13], with enacted stigma such as police interaction, homophobia, and discrimination due to gender identity often experienced.

In two drug classes (1) depressant and stimulant use, 2) high polysubstance use), opioid use was relatively high. This is concerning due to the ongoing opioid epidemic in the USA, together with overdose risk that is especially pertinent with SGMY who may face additional barriers to harm reduction and treatment services [40]. In addition, the study findings indicated that participants who identified as transgender were significantly more likely to report

opioid use, similar to previous literature with gender minority youth [17,41]. More specifically, recent studies have found that SGMY have increased odds of using non-medically prescribed prescription opioids [37,42,43], including transgender youth [44]. The increased risk of opioid use has previously been explained within the minority stress framework, as opioid use is viewed as a way to cope with increased life stressors [37,45], including discrimination [43] that GMY in particular may face. The findings therefore suggest that interventions with gender minority youth in particular should include assessment of opioid use including health education regarding the risks of mixing opioids and sedatives, and the provision of naloxone to prevent overdose deaths [46].

Arrest history was associated with all three typologies of polysubstance use, similar to previous studies with SGMY [47–49], which also found previous arrest predicts substance use [48]. The experience of prosecution among minority youth, especially Black or Hispanic youth, is of great concern [49] as this has been associated with other syndemic factors, which may lead to increased HIV risk behaviors such as participation in sex parties [50] and condomless anal sex [51]. The results also indicate that interventions provided to this population in criminal justice settings could be particularly effective, echoing previous findings [52]. It also indicates the need for multi-level intervention, such as police training to improve engagement with SGMY.

In terms of mental health, anxiety was associated with substance use in the current study. Participants in three of the categories of substance use (except low substance use class) were more likely to self-report increased anxiety levels. Previous studies have similarly found high levels of mental health symptoms among these minority groups [16], which have in turn been linked to substance use [53,54]. However, anxiety has generally been measured together with depression [7,55], making it difficult to disentangle these results. It is plausible that these youth engage in these patterns of substance use to manage high levels of anxiety stemming from minority stress [13], or that elevated anxiety symptoms are commensurate with withdrawal from substances. Further research is needed to understand whether mental health interventions targeting anxiety symptoms can assist sexual and gender minority youth with reducing polysubstance and other drug use.

Finally, participants in the high polysubstance use class were least likely to report lifetime HIV testing, despite recommendations around regular testing for minority youth who engage in high-risk behavior [56] to be status-aware and linked to the HIV continuum of care. This study seems to be one of a few that have examined engagement in substance use as a barrier to HIV testing [27,28]. Expanded efforts should be made to reach these youth in criminal justice settings, shelters, and community-based organizations to address substance use and facilitate HIV testing.

Study limitations need to be acknowledged. Class prevalence were unequal, with low sample sizes in both Classes 1 (depressant and stimulant use) and 2 (high polysubstance use). This unequal class sizes adversely affected the precision of our estimates, consequently leading to wider confidence intervals. Consequently, the generalizability of our results is limited, and the study's power is reduced. However, it is important to note that our decision was not solely based on class size. Instead, we followed Geiser's recommendation [57] to prioritize interpretability and goodness-of-fit measures when selecting the final class solution. Moreover, our classes, despite small sample sizes, contained different enough patterns of substance use to constitute distinct typologies. While it is important to note the high levels of substance use in some of these classes, it is also key to identify distinct patterns of substance use, as these may be associated with different levels of risk behaviors which may mean that health services may need to take these patterns into account accordingly. Nevertheless, to overcome these limitations, future studies should aim for increased statistical power and larger sample sizes. Such endeavors will address the limitations observed in our study and enhance the overall

understanding of the phenomenon. In addition, the majority of the sample was white, college-educated and lived in formal housing. It is possible that the online recruitment strategy was less successful in engaging 'hard-to-reach' groups of SGMY, such as those with high polysubstance use and other risk behaviors. Future research is warranted to measure whether the current substance use classes will be replicated. Although biomarkers for substance use through urinalysis were available later in the study, baseline measures were self-reported and may not be a true estimate of substance use prevalence. Recent studies found that both sexual and minority youth under-report substance use [2,17,23,58], which is influenced by certain demographic factors including ethnicity [59] and type of substance [48]. Finally, this study was restricted to Detroit, Michigan which may limit the generalizability of findings, and not be representative of substance use patterns among minority youth in the rest of the USA. Despite these limitations, the study findings add value by providing information on distinct patterns of polysubstance use in both sexual and gender minority youth.

## Conclusion

The current study findings expand on previous LCA studies that show patterns of polysubstance use among adult MSM and those who identify as part of a gender minority group, by focusing on a younger cohort of this high priority population. The study is also unique as it expands our knowledge on the relationship between polysubstance use typologies with specific mental and physical health outcomes, including lifetime HIV testing. Specifically, findings highlight the need to develop interventions for gender minority youth to target depressant use while mitigating overdose risk, deploy interventions targeting both sexual and gender minority youth in criminal justice settings to reach youth engaging in polysubstance use, and implement expanded efforts to promote HIV testing. Continued research into identifying distinct patterns of polysubstance use patterns will assist with developing targeted interventions to address health risks of substance use and optimize HIV/AIDS prevention with this high priority population.

## Supporting information

**S1 File. Data subset used for study analysis.**
(XLSX)

## Acknowledgments

The authors would like to thank the SWERVE research team, particularly Alexis Hunter, in providing detail on study activities needed to write this manuscript. We also thank all SWERVE participants that completed the online baseline survey.

## Author Contributions

**Conceptualization:** Tara Carney, Rob Stephenson, Jose A. Bauermeister, Adam W. Carrico.

**Formal analysis:** Seul Ki Choi.

**Funding acquisition:** Rob Stephenson, Jose A. Bauermeister.

**Methodology:** Adam W. Carrico.

**Writing – original draft:** Tara Carney, Seul Ki Choi.

**Writing – review & editing:** Tara Carney, Rob Stephenson, Jose A. Bauermeister, Adam W. Carrico.

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
