## [Decision Letter · Decision Letter 0]

15 Jun 2023

PONE-D-22-30812Latent class analysis of substance use typologies associated with mental and sexual health outcomes among gender identity and sexual orientation minority youthPLOS ONE

Dear Dr. Carney,

Thank you for submitting your manuscript to PLOS ONE. After careful consideration, we feel that it has merit but does not fully meet PLOS ONE’s publication criteria as it currently stands. Therefore, we invite you to submit a revised version of the manuscript that addresses the points raised during the review process.

As you can see, the reviewer's are overall pleased with the manuscript and suggestions are largely minor in nature and should be addressed easily. Please also see my specific comments below. 

We look forward to receiving your revised manuscript.

Kind regards,

Daniel Demant, PhD, MPH, GradCertHEd, BAppSocSc

Academic Editor

PLOS ONE

Additional Editor Comments:

The manuscript overall needs more focus and consistency in language. Please ensure to address issues related to latent class analysis in more detail in the manuscript, specifically the methodology.

Reviewers' comments:

Reviewer's Responses to Questions

**Comments to the Author**

1. Is the manuscript technically sound, and do the data support the conclusions?

Reviewer #1: Yes

Reviewer #2: Yes

2. Has the statistical analysis been performed appropriately and rigorously? 

Reviewer #1: No

Reviewer #2: Yes

3. Have the authors made all data underlying the findings in their manuscript fully available?

Reviewer #1: Yes

Reviewer #2: Yes

4. Is the manuscript presented in an intelligible fashion and written in standard English?

Reviewer #1: Yes

Reviewer #2: Yes

5. Review Comments to the Author

Reviewer #1: COMMENTS TO AUTHORS:

Overall comment: Thank for the opportunity to review this manuscript as it in an important area of work. However, there are a few issues with this paper that needs to be addressed. The paper also needs editing and proof-reading before submission.

Abstract

1. The statement “We applied latent class analysis…. (line 30-33)” needs rewording. In its current form it reads values on substance use variables were used to create latent class to predict substance use patterns. Please revise this.

2. There are two “=” signs when mentioning SD values.

Introduction

1. The introduction needs some work, and focus. The second line in first para (line 49, page 3), please ensure you are using the correct in-text citation when referencing to CDC surveillance, and rather than stating it as ‘recent’, please provide the year.

2. Similarly, please provide the year for YRBS survey (line 54, page 3).

3. There is also inconsistency in which the target population has been described throughout the paper; namely, young men who have sex with men (YMSM), sexual minority youth, sexual and gender minority, sexual and gender minority youth who have sex with men, sample of young men (cisgender and gender non-conforming), groups of youth, etc. Please revise the manuscript, and check for these inconsistencies. If it helps, operationalize it in the start of the paper, and explain what it means for this project.

4. I think the authors meant multiple forms or types of discrimination and not multiple discrimination indicating count of discrimination (line 66, page 3).

5. Page 4 (lines 70-77), it would be advisable to refer to minority stress theory/framework (Meyer, 2003), to contextualize the distal stressors as underlying mechanisms. It would also help to bind this para.

6. The references on specific STI’s are not needed (line 78, page 4), rather use references/citations on variables that are actually used in this paper. Otherwise, it sets readers to expect findings associated with STIs.

7. Page 5, line 95. Please revise the aim, it reads the purpose of this study is to better inform strategies; however, that is what is expected to happen from the results, but is not the main aim (and scope of this study).

Methods

1. Page 5, 101, again, the sample is referred to with another phrase “sample of young men who have sex with men, some of whom also identified as transgender or non-binary”. Please ensure consistency.

2. Page 7, line 137: authors have used ‘racial categories’ and ‘racial and ethnic categories’ in the same para. Please consistently use terms.

3. The methods section doesn’t state sample size for the study, for this particular paper (if not used the entire sample enrolled in this project).

4. The methods section also lacks a discussion on missingness in the variables and how missingness was handled in subsequent analyses.

Results/ Discussion/ Limitations

1. Can authors please clarify why a 4-class solution was selected (as stated by authors, BIC was higher for class 4 compared to class 3). I would also suggest adding -2Loglikelihhood and adjusted BIC (aBIC) values to the table. More importantly, my issue with the class selection is with the sample size in each class. Class 1 and class 2 have less than 5% of sample (n=14 (3.4%); and n=19 (4.6%); respectively). Classes 3 and 4, combined account for 92% of the sample. Most research suggests, to have at least 5% of the sample in each class; and with the smaller sample size in general, can authors clarify and discuss if the LCA regression analyses had enough power to detect differences.

2. The confidence intervals for some results are too wide, namely, in the unadjusted modes (page 13, line 269-276). Can authors please comment on this; and clarify this is not due to the power issue (enough samples in each class).

3. This discussion section sufficiently describes the findings and contextualizes these in the literature; however, it needs a focus and re-organizing. The authors should focus on discussing the major findings on key variables; and limit the discussion on control variables.

4. The authors may refer back to minority stress theory in the discussion, to discuss the findings.

5. If intervention strategies are one of the aims of this paper, then the authors must include a separate section, discuss what we currently have, and where interventions need to be adapted, or new interventions need to be developed.

6. Including, class sample proportions in the limitation section is not enough, the authors need to address this in the methods/results section; and even consider a 3-class solution.

Reviewer #2: Thank you for the opportunity to review this manuscript, which uses an LCA to determine typologies of substance use in YMSM as well as correlates of class membership. The article is interesting with good subject matter. I have a comments and suggested edits, more based on grammar than substance:

Line 49- “The recent HIV Behavior,” “The” should not be capitalized. Also, I believe it is called the “National HIV Behavioral Surveillance”

Line 61- I find this sentence a bit confusing: “who have also been found to have up to four times substance higher use than cisgender youth.” Do you mean they use substances at a rate 4 times higher than their cisgender peers? Or 4 times the substances? Please clarify.

Line 81- “Studies have found also found,” please edit to avoid duplicate language.

Line 95- There are a few Detroit’s in the US. It would be helpful to clarify that this is Detroit, Michigan.

Line 101- Sometimes you use YMSM, other times you spell it out. It’s good to be consistent and only use YMSM after you’ve defined your acronym the first time.

Line 142- “previous abuse by police” is interesting. How did you define that?

Line 174- “one third” should have a hyphen (one-third). So should two-thirds on line 176

Line 177- Please include a % for 63.8

Line 181- You have already defined PrEP on line 145, please only use the acronym from now on.

Line 262, 266- When you talk about controlling for other factors, what specific factors are you controlling for?

Line 305- Just a question for consideration, if you were looking at any cannabis in the last 3 months, but hazardous alcohol use, do you have any data on the prevalence of participants that may have hazardous cannabis use (CUDIT)? To me it seems any cannabis use vs. hazardous use if very different, same as any alcohol vs. hazardous alcohol use.

Line 308- were typologies compared to gender identity overall or transgender identify specifically?

Line 324- “In two drug classes, opioid use was relatively high.” Please name the classes for reader clarity, otherwise they have to go back and search.

Line 325- I find this sentence confusing, “together overdose risk that is especially pertinent with YMSM

youth that belong to who may face barriers to harm reduction and treatment services.” First, YMSM are youth so this is redundant. Second, what do you mean “youth that belong to who may face barriers”? I believe this sentence could use some edits for clarity.

Line 331- I find this sentence confusing, “The increased risk of opioid use among been explained within the minority stress framework.” I believe some edits could help to clarify meaning.

6. PLOS authors have the option to publish the peer review history of their article (what does this mean?). If published, this will include your full peer review and any attached files.

Reviewer #1: **Yes: **Ankur Srivastava

Reviewer #2: No

---

## [Author Response · Author response to Decision Letter 0]

2 Aug 2023

Please see letter attached that contains detailed information on responses to each reviewer.

---

## [Editor Report · Decision Letter 1]

16 Aug 2023

Latent class analysis of substance use typologies associated with mental and sexual health outcomes among sexual and gender minority youth

PONE-D-22-30812R1

Dear Dr. Carney,

We’re pleased to inform you that your manuscript has been judged scientifically suitable for publication and will be formally accepted for publication once it meets all outstanding technical requirements.

Kind regards,

Daniel Demant, PhD, MPH, GradCertHEd, BAppSocSc

Academic Editor

PLOS ONE

Additional Editor Comments (optional):

The authors have addressed the feedback sufficiently.
---

## [Editor Report · Acceptance letter]

19 Sep 2023

PONE-D-22-30812R1 

Latent class analysis of substance use typologies associated with mental and sexual health outcomes among sexual and gender minority youth   

Dear Dr. Carney:

I'm pleased to inform you that your manuscript has been deemed suitable for publication in PLOS ONE. Congratulations! Your manuscript is now with our production department. 

Kind regards, 

on behalf of

Dr. Daniel Demant 

Academic Editor

PLOS ONE